# Immediate Effects of Sforzesco^®^ Bracing on Respiratory Function in Adolescents with Idiopathic Scoliosis

**DOI:** 10.3390/healthcare9101372

**Published:** 2021-10-14

**Authors:** Fabrizio Di Maria, Andrea Vescio, Alessia Caldaci, Ada Vancheri, Chiara Di Maria, Marco Sapienza, Gianluca Testa, Vito Pavone

**Affiliations:** 1Section of Orthopaedics and Traumatology, Department of General Surgery and Medical Surgical Specialties, University Hospital Policlinico “Rodolico-San Marco”, University of Catania, 95123 Catania, Italy; fdimaria95@gmail.com (F.D.M.); alessia.c.92@hotmail.it (A.C.); marcosapienza09@yahoo.it (M.S.); gianpavel@hotmail.com (G.T.); 2Department of Clinical and Experimental Medicine, Regional Referral Center for Rare Lung Disease, University of Catania, 95123 Catania, Italy; adact1@hotmail.it (A.V.); chiaradima@hotmail.com (C.D.M.)

**Keywords:** adolescent idiopathic scoliosis, Sforzesco brace, pulmonary function

## Abstract

The thoraco-lumbar bracing is an effective management of adolescent idiopathic scoliosis (AIS). Studies have shown that brace wearing reduces lung volume. Whether or not the Sforzesco brace, frequently used in Italy, affects lung volume has not been investigated. We studied the immediate effect of Sforzesco bracing on lung volumes in 11 AIS patients (10 F, 1 M; aged 13.6 ± 1.6 yrs) mean Cobb angle 26 ± 4.49 degrees. Lung function variables and the perceived respiratory effort were recorded twice, before and 5 min after bracing. The one-way analysis of variance repeated measures, and multiple comparison tests, showed that means of unbraced variables were not significantly different from the corresponding means of predicted values, whereas means under brace were significantly lower (*p* < 0.05) compared to both predicted and baseline values of respiratory variables. In addition, a significant correlation (*p* < 0.0001) was found between unbraced and braced values, and linear regression equations were calculated. A significant but clinically unimportant increase in perceived effort was observed under the brace. In conclusion, data indicate that lung function is not impaired in moderate AIS and that wearing the Sforzesco brace causes an immediate, predictable reduction of lung volumes. Data also suggest that the respiratory discomfort during brace wearing could not be due to respiratory function defects.

## 1. Introduction

Idiopathic scoliosis is characterized by a lateral curvature of the spine, with a Cobb angle of more than 10 degrees and rotation of the vertebral column along its longitudinal axis [1]. Due to skeletal immaturity, adolescent patients are at risk of curvature progression through adulthood, causing chest deformity and movement abnormalities of the rib cage, thereby reducing chest compliance and lung volume [2].

During the past few decades, several studies have confirmed that adolescent idiopathic scoliosis (AIS) can be positively affected by wearing a rigid thoracolumbar brace [3]. There are many brace designs, all acting through external compressive forces on the chest wall to stop or reduce the progression of spinal curvature, with more hours of bracing associated with greater benefit [3].

Similar to chest wall strapping that reduces thoracic distensibility and restricts lung volumes in normal subjects [4], brace wearing impairs respiratory mechanics [5,6] and results in a reduction of pulmonary function [7,8] and increased dyspnea perception [8].

Respiratory discomfort may contribute to limiting the patient’s adherence to using it, thus reducing its long-term efficacy.

The Sforzesco brace, one of the most used in Italy, is a custom-fitted thoraco-lumbar orthosis, manufactured according to Symmetric, a patient-oriented, rigid, three-dimensional, active (SPoRT) concept [9,10]. Its particular design allows for active correction, as the patient moves the spine from the pressures generated within the brace, although its acute respiratory effects have not yet been studied.

The aim of this study is to determine whether baseline respiratory function parameters, compared to their predicted values, were altered, and whether static and dynamic lung volume and perceived respiratory effort are affected by the Sforzesco brace in a group of adolescents with moderate idiopathic scoliosis.

## 2. Materials and Methods

Eleven subjects (1 male, 10 females, aged 11–16 years) with moderate AIS, out of 127 consecutive patients attending outpatient clinics in the Section of Orthopaedics at the University Hospital Policlinico-San Marco, Catania, Italy, were enrolled between April 2018 and May 2019.

Inclusion criteria were: (1) confirmed diagnosis of moderate AIS (Cobb angle of 21° to 35° for the primary curve); (2) a thoracic or thoracolumbar primary curve; (3) skeletal immaturity with growth cartilage visible on pre-treatment radiographs (Risser score < 5); (4) age of 11 to 16 years; and (5) ongoing treatment with the Sforzesco brace. Exclusion criteria were: (1) scoliosis due to other spine disorders, or linked to known causes; (2) present or past smoking habit; (3) presence of cardio-respiratory, pulmonary and/or pleural abnormalities; (4) upper or lower respiratory tract infections in the 6 week period before the study day.

Demographic and relevant clinical data were recorded. The Cobb angle and Risser classification were assessed by the same senior orthopedic surgeon (V.P.) according to Scoliosis Research Society guidelines [11]. To make up for the height loss resulting from spine curvature, individual statures were figured out by the Armspan-to-Height Software [12] and used to calculate predicted values of lung function parameters [13].

The patient’s BMI and BMI for-age z-score were also calculated, using the WHO AnthroPlus software, Copenhagen, DK [14]. Vital statistics and other relevant characteristics of enrolled patients are shown in Table 1.

Upon enrollment, an informed consent stating the purpose of the study and the possibility of anonymously publishing the data, approved by the hospital Ethics Committee, was obtained for all study subjects from at least one of their parents or legal representatives. Patients were also asked to refrain from wearing their brace 24 h before the study day to avoid carry-over effects of current brace treatment on study measurements.

Vital capacity (VC), residual volume (RV), total lung capacity (TLC), forced expiratory volume in 1 s (FEV_1_), forced vital capacity (FVC), FEV1/FVC ratio, intrathoracic gas volume (ITGV), peak expiratory flow, and maximal voluntary ventilation (MVV) were measured.

Respiratory function was assessed while sitting, using a calibrated body plethysmograph (MasterScreen Body Jaeger, GY), in accordance with the joint American Thoracic Society and European Respiratory Society standards [15]. Lung volumes were obtained by electronic integration of mouth airflow, as measured by a Fleisch #3 pneumotachograph and corrected to BTPS (body temperature and pressure standards). The best of three reproducible measured values was used for analysis.

Respiratory discomfort as a sensation of inspiratory effort was assessed by asking subjects to take a deep breath followed by exhalation, and then asking them to rate their perceived effort on a modified 0–10 Borg scale [16].

Measurements were obtained after establishing a resting breathing before and 5 min after wearing the Sforzesco thoracolumbar brace.

All data were expressed as measured absolute values and predicted values by using reference data of the Global Lung Initiative [17].

The one-way analysis of variance (ANOVA) for repeated measures and the Tukey’s multiple comparison test were used to compare the means of pulmonary function measurements and the absolute predicted values. The correlation between the patient’s BMI-for the age z-score and Borg rating of perceived respiratory effort was determined using the linear regression. The t-test was also used where appropriate. The rule-of-thumb scale to evaluate the correlation coefficient was used: for each respiratory parameter, linear regression analysis was also used. All statistical analyses were performed with the computing package, GraphPad Prism Version 5.0 (GraphPad Software Inc., San Diego, CA, USA). Continuous data are presented as means and standard error of the means (SEM) or standard deviation (SD) as appropriate. The selected threshold for statistical significance was *p* < 0.05.

## 3. Results

The means of all studied pulmonary function variables and results of one-way analysis of variance for repeated measures, and multiple comparison test, are presented in Table 2.

Analysis of variance indicated there were differences among the means of the three data groups, i.e., measurements before and with bracing, and the corresponding predicted values. Differences were significant (*p* < 0.0001) for VC, TLC, ITGV, FEV1, FVC, and PEFR, but not for RV, the FEV1/FVC ratio, and MVV. The means of baseline respiratory parameters were slightly lower compared to predicted values, and the multiple comparison Tukey’s test consistently showed that differences were not significant, except for FVC and PEFR (*p* < 0.05; Table 2). We also found that, compared to baseline, respiratory parameters with bracing were significantly lower (*p* < 0.05)—along with a corresponding mean reduction in brackets—for VC (−22%), TLC (−14%), ITGV (−21%), FEV1 (−21%), and FVC (−19%), but not for RV, FEV1/FVC ratio, MVV, and PEFR.

Pulmonary function variables showed a highly significant linear correlation between baseline and bracing (*p* < 0.0001, except for the FEV1/FVC ratio (*p* = 0.0054)). Scatter diagrams and corresponding regression lines are reported in Figure 1.

Bracing was associated with an increased perception of respiratory effort in some, but not all subjects. The mean (±SEM) Borg score increased from 0.09 ± 0.06 at baseline to 0.91 ± 0.36 under brace (*t*-test, *p* = 0.011). Finally, by using the linear regression analysis, a significant inverse correlation (r = −0.68; *p* = 0.021) was found between the BMI-for-age z-score and the perceived respiratory effort (Figure 2), but not any of the percent reduction of respiratory variables.

## 4. Discussion

This study shows that baseline ventilatory function of adolescent patients with moderate idiopathic scoliosis under treatment with orthotic thoraco-lumbar bracing is not altered. The study also demonstrates that the application of the Sforzesco brace causes a significant decrease in lung function, which in some patients is associated with increased perception of respiratory effort. In addition, results show that the higher the BMI, the lower the perception of respiratory effort associated with brace application. Finally, and more interestingly, we found that lung function values obtained before and during bracing are highly correlated, and that the immediate reduction of respiratory function from brace wearing can be predicted by using simple linear regression equations.

The impact of moderate scoliosis on pulmonary function is less evident, with patients having no or few respiratory disturbances and being clinically asymptomatic [18,19]. This is confirmed by our data, showing that patients with moderate scoliosis and an average Cobb angle of 26° have no respiratory discomfort at rest and no or minor insignificant pulmonary function alterations. The absence of lung function impairment along with no or negligible respiratory discomfort at baseline is similar to that reported by other studies [7]. Differences are not significant except for FVC and PEFR (*p* < 0.05).

Bracing has been shown to be a successful treatment option for patients with AIS [20]. In a multicenter study, it significantly decreased progression of high-risk spine curvature to the threshold of surgery with longer duration of brace wearing associated with greater benefit [3]. Given current evidence, bracing has been incorporated in the guidelines and recommended as a first line treatment for management of AIS in patients with a Cobb angle above 20°, or more likely between 20° and 40° [11,21]. Despite long-term efficacy of bracing treatment, several authors highlighted the possible disadvantages, including a short-term decrease in pulmonary function and maximal exercise performance limitations, especially in girls [5,6,7,8].

One study described an immediate, significant, and uniform decrease in VC, RV, functional residual capacity (FRC), TLC, and FEV1, with unaltered FEV1/VC ratio in a group of patients affected by thoracic or thoracolumbar AIS, treated with the Boston brace [7]. Similar results in a comparable group of patients were obtained after the Milwaukee and Boston brace application [5]. Recently, a study investigating the acute effect of a cTLSO (custom-made rigid thoracolumbosacral orthosis) bracing treatment in moderate AIS found a significant reduction of pulmonary function parameters, including VC, FEV1, FVC, MVV, and PEF, along with an increased perception of dyspnea [8].

These results are not trivial, given the reduction of respiratory parameters and the occurrence of dyspnea due to brace wearing—which may contribute to poor adherence of the long-term brace treatment as observed in many patients.

It is generally agreed that lung function abnormalities from brace wearing are mainly the restrictive type. This is similarly true for our study. It should be noted, however, that we found a clear-cut reduction of TLC but not RV. Although the interpretation of differences with bracing is beyond the purpose of our study, we can speculate that a possible explanation is that airflow obstruction could occur due to decreased efficacy of respiratory muscles with adolescent patients using the brace.

Although significant changes in pulmonary function values, induced by bracing, were found in all patients for most respiratory parameters, a clinically important increase in dyspnea was observed in only two of eleven patients, while negligible or no increase was found in four and five patients, respectively. This finding suggests that dyspnea reported by some patients under these circumstances could not be dependent on the occurrence of respiratory function alterations, and that other factors including chest wall constriction per se or psychological distress, might explain the perceived respiratory discomfort. The observation that reduced respiratory function is not or poorly perceived may be important, as far as it suggests it does not represent a limit to treatment adherence. The rate of successful brace treatment is largely related to the average hours of daily brace wear and the continuity of brace treatment.

We wish to also emphasize that the high degree of correlation observed for all measured lung function parameters, between unbraced and braced conditions in AIS patients and the subsequent predictability of changes in respiratory function immediately after brace wearing may be helpful in clinical practice.

Finally, the negative correlation between BMI-for-age z-score and Borg rating suggests that the higher BMI might mitigate the respiratory discomfort perceived under brace. This is conflicting with the evidence that some obese but otherwise healthy subjects experience dyspnea at rest [22]. Unfortunately, the small sample size and the unbalanced gender groups in our study do not allow us to draw firm conclusions about this important aspect.

## 5. Conclusions

The immediate effect of the Sforzesco brace on pulmonary function of patients with moderate AIS had not, so far, been investigated.

The present study demonstrates that baseline lung volumes compared to their predictive values are slightly reduced in our AIS patients, but reduction doesn’t reach statistical significance. Wearing the Sforzesco brace is immediately followed by a significant, and rather uniform reduction in lung volume.

The findings of present study confirm that patients with moderate AIS have a preserved baseline lung function and are asymptomatic. Instead, the application of the Sforzesco brace is associated with a significant decrease in lung function—which in some patients is accompanied by an increased perception of respiratory effort. Regression analysis showed that the immediate response of respiratory variables to brace wearing is highly predictable. In addition, results allow us to postulate that a higher body weight could mitigate the intensity of respiratory effort perceived upon application of Sforzesco brace. 

Further studies are needed to check and expand the results of the present study, as well as to explore the effect of bracing on exercise, quality of life, and treatment adherence in adolescent patients with moderate idiopathic scoliosis.

## Figures and Tables

**Figure 1 healthcare-09-01372-f001:**
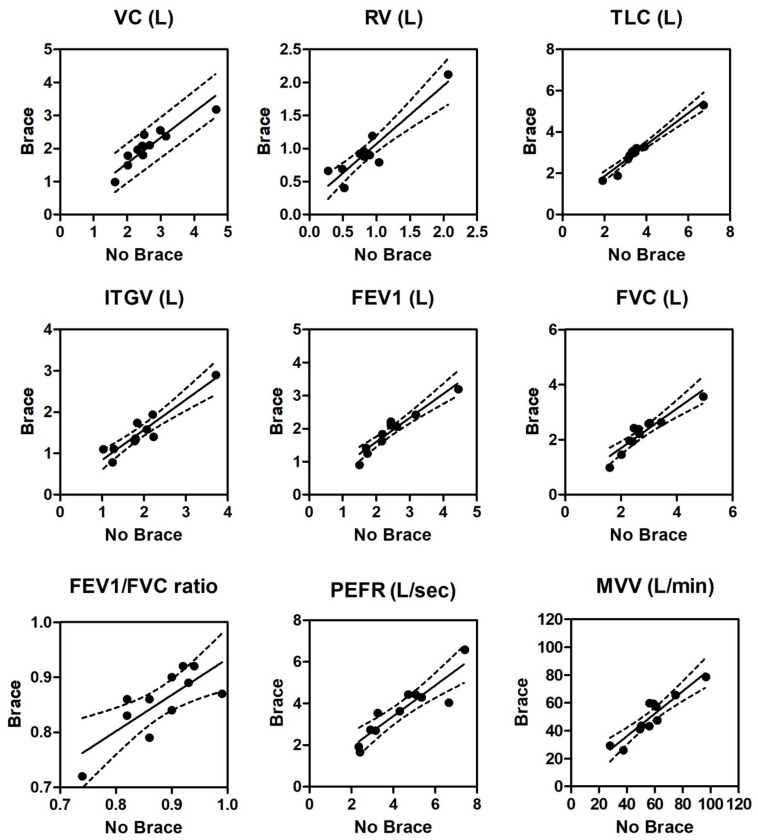
Scatter diagrams and regression lines (dashed lines are 95% CIs) for lung volumes and other respiratory parameters at baseline (x axis) and immediately after wearing the Sforzeso thoracolumbar brace (y axis). VC is vital capacity, RV residual volume, TLC total lung capacity, ITGV intrathoracic gas volume, FEV1 forced expiratory volume for 1 s, FVC forced vital capacity, FEV1/FVC ratio, PEFR peak expiratory flow rate, and MVV maximal voluntary ventilation. Correlation was significant at *p* < 0.0001 for each set of measurements, except FEV1/FVC ratio (*p* = 0.0054). Correlation coefficients and regression equations for each parameter set are seen in the Appendix A (Table A1).

**Figure 2 healthcare-09-01372-f002:**
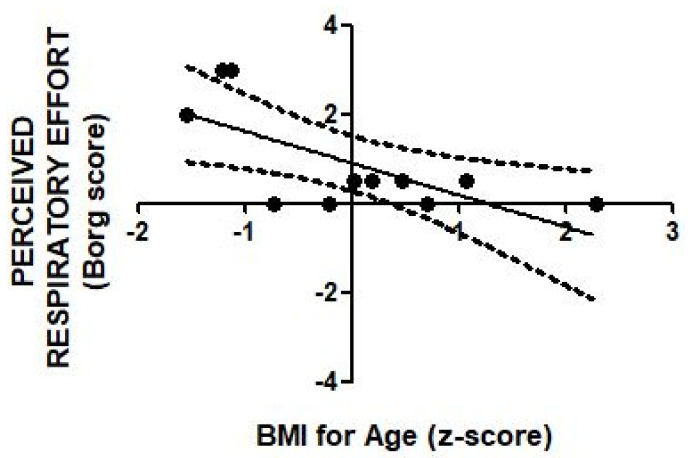
Scatter diagram of individual values of BMI-for-age z-score and the Borg rating of perceived respiratory effort in adolescent scoliosis with the Sforzesco brace. The solid line depicts regression function of y on x; the inverse correlation is significant (r = −0.68; *p* = 0.021). Dashed curves depict 95% confidence intervals.

**Table 1 healthcare-09-01372-t001:** Vital statistics of study subjects. BMI = body mass index; F = female; M = male; SD = standard deviation; Min = minimum: Max = maximum. Note that arm-span was used as a surrogate of height.

	Patient Initials	Age (yrs)	Gender	Height (cm)	Weight (kg)	BMI-for-Age z-Score	Time of Treatment (Month)	Cobb Angle (Grade)	Risser Score
1	M.A.P.	12	F	129	28.3	0.19	18	30	0
2	D.A.	15	F	145	49.1	1.07	8	22	3
3	C.S.	11	F	158	45.0	0.47	12	23	0
4	G.S.M.	13	F	151	47.4	0.70	19	34	3
5	C.M.G.	14	F	152	46.3	0.03	10	24	3
6	P.A.	12	F	121	22.3	−0.21	1	22	0
7	L.M.	15	M	178	53.6	−1.54	1	22	3
8	L.R.	16	F	162	47.2	−1.21	24	30	4
9	C.B.	13	F	157	43.0	−0.72	2	28	0
10	L.A.	12	F	140	43.5	2.29	15	30	0
11	C.M.	13	F	146	35.1	−1.12	1	21	0
Mean		13.27		149	41.9	−0.0045	10.09	26	1.45
±SD		1.56		15.68	9.46	1.13	8.25	4.49	1.69
Min		11		121	22.3	−1.54	1	21	0
Max		16		178	53.6	2.29	24	34	4

**Table 2 healthcare-09-01372-t002:** Pulmonary function parameters are presented as mean ± standard error of the mean (SEM). The *p*-values are based on a 1-way ANOVA test for repeated measures and (*) Tukey’s multiple comparison test. VC: vital capacity; RV: residual volume; TLC: total lung capacity; ITGV: intrathoracic gas volume; FEV1: forced expiratory volume in the first second; FVC: forced vital capacity; MVV: maximal voluntary ventilation; PEFR: peak expiratory flow rate.

	VC (L)	RV (L)	TLC (L)	ITGV (L)	FEV1 (L)	FVC (L)	FEV1/FVC Ratio	MVV L/min	PEFR L/sec
Predicted	2.79 (0.26)	0.95 (0.08)	3.70 (0.31)	1.84 (0.17)	2.52 (0.23)	2.85 (0.92)	0.88 (0.00)	50.12 (4.74)	5.12 (0.39)
No Brace	2.63 (0.24)	0.86 (0.14)	3.52 (0.36)	1.91 (0.22)	2.44 (0.25)	2.77 (0.88)	0.88 (0.02)	57.38 (5.44)	4.32 (0.51)
Brace	2.06 (0.17)	0.95 (0.13)	3.02 (0.28)	1.50 (0.17)	1.92 (0.19)	2.25 (0.68)	0.85 (0.02)	50.13 (4.75)	3.63 (0.42)
1-way ANOVA	*p* < 0.0001	NS	*p* < 0.0001	*p* < 0.0001	*p* < 0.0001	*p* < 0.0001	NS	NS	*p* < 0.0001
F-value	29.21	1.21	18.50	14.75	22.04	34.56	1.05	2.29	15.03
Pred vs. No Brace *	NS	NS	NS	NS	NS	*p* < 0.05	NS	NS	*p* < 0.05
Pred vs. Brace *	*p* < 0.05	NS	*p* < 0.05	*p* < 0.05	*p* < 0.05	*p* < 0.05	NS	NS	*p* < 0.05
No Brace vs. Brace *	*p* < 0.05	NS	*p* < 0.05	*p* < 0.05	*p* < 0.05	*p* < 0.05	NS	NS	NS

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
