# Peer review of "Immediate Effects of Sforzesco® Bracing on Respiratory Function in Adolescents with Idiopathic Scoliosis"

_healthcare, 2021, doi:10.3390/healthcare9101372_

Round 1

Reviewer 1 Report

General impression

In this article, the authors investigated the effects of on chest wall mechanics and lung function in patients with adolescent idiopathic scoliosis (AIS) who treated with the Sforzesco brace.  And they concluded that the Sforzesco brace in AIS patients results in a transient, uniform, and predictable reduction of pulmonary function on the basis of their results.  Although the number of cases is quite small, I believe the information in this study must be valuable for the physicians to manage AIS as a preliminary report.

The methodology of this study was precisely explained and acceptable.  Also, the limitations of this project were well indicated.  For these reasons, I think this manuscript is appropriate for publication.

However, I have a couple of minor requests to be revised as stated below.  After they have been resolved, I will judge this manuscript can be accepted and published by the healthcare journal.

  1. Abstract line 19

 I guess “alter” should be replaced by “after”.

  1. abbreviation of “Pred”: page 2 Table 2

  I assume “Pred” must be an abbreviation of “predicted value”.  However, some readers may not understand it.  I recommend the total words should be indicated at the title and legend.

Author Response

1.Abstract, line 19: “alter” has been changed to “after”.

2.Page2, Table 2: “Pred” has been changed to “Predicted”.

Reviewer 2 Report

The AIS patients only have a small population, however, their suffering and discomfort are last lifelong, therefore this study findings have given further investigation to suggest.

This is a well-written paper, the research designs to fit criteria in many aspects, through the cases number are rather small, however, AIS cases could be difficult to meet the selective sample criteria.

In general, the reviewer only has one question that needs to be addressed, that in this study a total of 4 cases under treatment only one-two months compared to the rest of 7 cases have much longer bracing treatment, whether the sensation of being used the bracing treatment would causing the result differently? please explain in further detail in the paper

Author Response

Answer to the reviewer’s comment on different duration of brace treatment among study subjects:

There is no putative reason for a difference in the immediate response to brace wearing amenable to the duration of previous brace treatment. As already mentioned in the paper, however, to avoid possible carry-over effect we asked subjects to refrain from wearing their brace 24 hrs before study (see Methods section of manuscript). Our results suggest lack of bracing influence inasmuch as changes in lung function due to brace turned out to be sufficiently predictable in each individual subject. Finally, one referenced article (Kennedy et al., Thorax  1987) achieving similar magnitude and distribution of lung function changes  included subjects with brace duration ranging from 0.1 to 5.0 yrs.

Reviewer 3 Report

General comments

This is a small study on the immediate effects (5 minutes after) of wearing a specific corrective orthotic bracing on pulmonary function and perceived respiratory effort at rest as measured by plethysmography and Borg scale, respectively. The research design is relatively simple and straightforward; however, I have major concerns regarding the analysis, which I detail later.

General recommendations and concerns:

  • Acute is most often used to describe responses up to 72h. Given their research design and measurements, I suggest that authors replace “acute” by “immediate” throughout the manuscript when referring to their study.
  • Apparently, ethical clearance for conducting this research has not been obtained. There is a difference (important) between assessing the effects of interventions in clinical practice and assessing the effects of interventions in clinical research. At least, patients and their representatives must know that they are in a study and that their data, even if anonymous, will be used for scientific purposes (e.g., publications). This is even more important because we are dealing with a vulnerable population (children), even if the intervention is non-invasive. This study should have been submitted to an ethical committee and their approval obtained.

Specific comments:

Title: See my general recommendations

Abstract:

I’m not sure if most readers understand the acronyms of pulmonary function metrics

Line 22: The authors conclude that the wearing of the brace results in a “transient” change in pulmonary function, however, their research design does not allow this conclusion because only one moment after the intervention was assessed. Please, amend.

Introduction:

Line 26: “of the lateral curvature” sounds strange. Can the authors please reword?

Line 37: Can the authors give some examples of the acute changes in respiratory mechanics and the link between those changes and respiratory discomfort? This is one important part of the rationale for having conducted this research but is somewhat superficial.

Line 39: Inform readers the indications of this brace (e.g., type of curvature, range of the magnitude of the spinal deformation, age ranges, etc.) so that they can later understand your eligibility criteria, in the Methods section.

Materials and Methods:

Line 61: Please inform if measurements were performed in film or in a digital format. If digital, inform about the software used.

Line 62: Please inform how height and weight (and instruments) were measured.

Lines 88–89: Can the authors please inform which ANOVA model have they used? According to Results section, One-way ANOVA has been used, but this is not correct because you have only one group and two sources (or factors) of within-subject variance: time (before and after 5 minutes) and “condition” (real measurement versus predicted). Interaction between these two factors is the most interesting approach to analyse. You cannot treat Predict, Brace and No-brace as different groups as it will inflate the variance and p-value.

Lines 90–91: Not sure why are authors are testing this. This has not been set as a study objective and if this was used to assess whether “…perceived respiratory effort are affected by the Sforzesco brace…” correlation coefficients are not inference techniques. Also, I’m familiar with Spearman’s rho and Pearson’s r correlation coefficients. This one is new for me. Can the authors please explain their rationale and this correlation coefficient?

Line 94: Please indicate which version of GraphPad was used.

Results:

I’m not confident with these Results because of unclear ANOVA model used.

Line 124: Why a t-test appears here when only ANOVA has been planned for statistical inference?

Readers are not prepared to appraise the Results on the association between pulmonary function metrics and wearing/not wearing brace because they have not been introduced to it. This looks like exploratory analysis but justification for using it have not been provided.

Discussion:

Line 144: The authors cannot state “…transient decrease in lung function,…” because you don’t have a follow-up, just the immediate effects. Please, revise.

Line 189: the term “disruption” seems a little bit dramatic. I suggest using “distress” instead.

Lines 198–200: The authors have just studied pulmonary function at rest. Pulmonary function at rest cannot predict pulmonary function during effort/physical activity. I suggest this as a limitation as well. How have your results been affected by the small sample size?

Conclusion:

Lines 203–207: This has not been studied here. I suggest removing this part and move it to Discussion.

Line 211: See my past comments concerning the term “transient”

Final sentence: The rationale for conducting this research was the potential association between a change in pulmonary function, respiratory discomfort, and long-term adherence to corrective bracing therapy. Why are safety aspects appearing here?

Author Response

We are particularly grateful for your revision, that let us to clarify and improve some important aspects and, hopefully, ameliorate the quality of our paper.

Point-by-point answers:

The SOSORT Guidelines (Negrini et al., Scoliosis and Spinal Disorders 2018) state that due to altered chest wall mechanics adolescent idiopathic scoliosis (AIS) can affect the respiratory system, and recommend that braces do not so restrict thorax excursion in a way that reduces lung function.  Thus the periodic assessment of lung function is routinely done in our AIS patients undergoing brace treatment and this practice is included in clinical protocols of our hospital. Therefore, it was sufficient to inform the hospital ethics committee which approved the content of written informed consent. Patients and their representative were adequately informed about the purpose of our “simple and straightforward study” as well as about the possibility of including anonymously in a scientific publication the data obtained in that study day.

Title: “Acute” replaced with “Immediate”

Abstract:

Acronyms of pulmonary function metrics have been removed from the abstract.

The word “transient” has been removed throughout the manuscript.

Introduction:

The phrase containing “…of the lateral curvature” has been reworded.

The link of “acute” changes in respiratory mechanics is well known and exemplified by restrictive changes and the subsequent onset of dyspnea or some of its qualitative dimensions, such as the sensation of unsatisfied inspiratory effort or shallow breathing induced by chest wall strapping in normal individuals. All these responses, which are particularly evident during exercise, may reasonably be attributed to the diminished ability to expand the thorax appropriately and to generate changes in lung volumes (O’Donnell et al. Respiratory sensation during chest wall restriction and dead space loading in exercising men. J Appl Physiol 2000).

There are papers and publications explaining indications of braces in general as well as the features of specific brace models. We provided suitable references for these important aspects but we also think that only topics that are closely relevant for the purpose of the study should be mentioned in detail. Otherwise, we could be called to deal with an endless list of topics.

Materials and Methods:

Spirometry measurements were obtained by the built-in software included in the body-box of Jager manufacturer.

Body weight was assessed by a calibrated scale like those that are usually present in the laboratories of respiratory physiology. Patients were barefoot with underwear. Height in cm was measured standing and barefoot by the anthropometric rod in the laboratory. Arm-span measured by stretching arms of patients while standing erect was used as an estimate of height.

Which method of analysis is appropriate depends on what questions the investigators ask. We asked whether the three means of each respiratory variable for just one source (or factor) of within-subject variance i.e. the ‘condition’ (predicted, unbraced, and braced) were different [contrary to your point of view, time is not a source of variance in this context]. Differences were found among the three group means of a number of variables, but since this analysis does not pinpoint which groups differs from the others, we used the multiple comparison Tukey’s test in order to know: (1) whether baseline lung volumes in AIS were reduced compared to their predicted normal values and (2) whether there was a brace-associated lung volume reduction in comparison with the unbraced condition.  The one-way repeated measures ANOVA (also known as a within-subjects ANOVA) is used to determine whether three or more group means are different where the participants are the same in each group (Medical Uses of Statistics 2nd ed. Bailar III JC & Mosteller F Eds. NEJM Book, Boston, MA, 1992). It is clear that here “group means” refers to a group of measures rather than a group of patients. Since this is widely recognized, we didn’t mention it in the statistics paragraph. For clarity, however, we rephrased “The analysis of variance (ANOVA)” by “The one-way analysis of variance for repeated measures”, both in the abstract and methods section of the manuscript.

For any given respiratory variable we compared the means of three “conditions”: predicted, unbraced, and braced (there is no time factor in our study). In our opinion, by treating this three “conditions” as data means within the same group of subjects is even more conservative and by weakening the signal of variability supports our results better. Hence, the “inflation” of both variance and P-value supposedly attributed to our statistical approach is, in our opinion, disproved by the results of multiple comparison showing that means are significantly different between unbraced and braced condition, and predicted versus brace, but not between predicted ad unbraced.

Lines 90-91: According to current evidence, chest wall strapping induces breathing at low lung volumes and is associated with a greater dyspnea intensity at any given workload (…). Mild to moderate obesity, due to chest wall and abdominal restriction, can also lead to reduction in lung volumes and dyspnea (…). In our study, AIS patients had volume reduction and experienced a significant increase in the perceived respiratory effort while wearing their brace. Therefore, we reasoned that ….and decided to explore whether the respiratory effort perceived under brace was associated to the BMI z-score. Surprisingly, we found an inverse significant correlation between the BMI z-score and the perception of respiratory effort. This suggests that overweight mitigates the brace-associated respiratory discomfort.

The Reviewer is perfectly right, …Pearson’s rho is wrong and maybe due to an oversight in that we assessed the relationship between BMI z-score and perceived effort by using the linear regression analysis. The text has been amended accordingly.

Lines 94: GraphPad version 5.0, text has been amended.

Results:

We hope we have cleared our statistical approach.

We have have included t-test in the methods section.

We stated – and adequately referenced – the association between bracing and lung function reduction and declared the purpose of our study in the introduction (paragraphs 3 to 5). Consequently, readers  should be sufficiently prepared to appraise the results. However, to comply with the Reviewer’s request, we added one referenced statement in the introduction (para 3).

Discussion:

Line 144: “transient” has been removed.

Line 189: “disruption” has been changed to “distress”.

The Reviewer’s observation headed “Lines 198-200” is maybe originated by a misunderstanding. We never discussed the possibility of predicting pulmonary function on exercise. We are not sure if the small sample size may have affected our results.

Conclusion:

Line 203-207: This para has been removed

Line 211: “transient” has been removed.

Round 2

Reviewer 3 Report

In general, I’m satisfied with authors clarifications amendments performed in this version of the manuscript.

A few suggestions to improve the quality of reporting though.

Line 238 – until it could be further studied, I suggest replacing “weight excess” by “higher BMI”, the latter already used by the authors in the previous version of the manuscript.

Maybe the authors have misunderstood my comment/suggestion regarding the limitations of studying pulmonary function and discomfort at rest (what the authors measured) and extrapolating them to what happens to pulmonary function (and discomfort) during exercise or physical activity (adolescents are supposed to move to have a fulfilling life, including physical education classes). This is well known in cardiovascular and pulmonary diseases (1), but also recognized in adolescent scoliosis (2). At this point, the findings of this study can’t be extrapolated to situations where patients using the Sforzesco Bracing are exercising or in (moderate to intense) physical activities. There is reasonable doubt that when pulmonary and cardiovascular function demands substantially increase (e.g. during vigorous exercise), there may be ventilatory impairments (and higher discomfort). Not having studied pulmonary function and discomfort during exercise (many adolescents are or should be physically active) could be regarded as both a study limitation and a recommendation for future research. I think the authors should include this in their study limitations section.

1 – ERS Task Force, Palange P, Ward SA, et al. Recommendations on the use of exercise testing in clinical practice. Eur Respir J. 2007;29(1):185-209. doi:10.1183/09031936.00046906

2 – Shen J, Lin Y, Luo J, Xiao Y. Cardiopulmonary Exercise Testing in Patients with Idiopathic Scoliosis. J Bone Joint Surg Am. 2016;98(19):1614-1622. doi:10.2106/JBJS.15.01403

Author Response

1. Discussion. Line 238: “weight excess” has been changed to “higher BMI”

2. Conclusion. We revised our conclusion. However, we think that point is not a limitation, because is not a purpose of our study.